# Learning Loss for Test-Time Augmentation

**Ildoo Kim**[*]
Kakao Brain
ildoo.kim@kakaobrain.com

**Younghoon Kim**[*][†]
Sungshin Women's University
yhkim@sungshin.ac.kr

**Sungwoong Kim**
Kakao Brain
swkim@kakaobrain.com

## Abstract

Data augmentation has been actively studied for robust neural networks. Most of the recent data augmentation methods focus on augmenting datasets during the training phase. At the testing phase, simple transformations are still widely used for test-time augmentation. This paper proposes a novel instance-level test-time augmentation that efficiently selects suitable transformations for a test input. Our proposed method involves an auxiliary module to predict the loss of each possible transformation given the input. Then, the transformations having lower predicted losses are applied to the input. The network obtains the results by averaging the prediction results of augmented inputs. Experimental results on several image classification benchmarks show that the proposed instance-aware test-time augmentation improves the model's robustness against various corruptions.

## 1 Introduction

Various autonomous systems (e.g., autonomous vehicle [11], medical diagnosis [2, 58], fault detection in the manufacturing process [27]) try to adopt neural networks as visual recognition module. The neural networks efficiently learn visual patterns to classify critical objects such as humans on the roads, cancers in our body, and manufacturing products' faults. Although recent research on deep learning with the benchmark datasets has shown promising results [5, 44], robustness problems can arise in real-world applications. As shown in previous works [19, 20, 15], the classification result can be easily broken even with slight deformations to the input image. In processing an input image using neural networks in real-world situations, several variations or corruptions can occur, leading to unexpected results [38]. Ensuring robustness is mission-critical in many applications, so many researchers have focused on the problem of neural networks [16, 51, 35, 3, 39, 31].

Recently, advanced data augmentation techniques have been proposed to improve the robustness of neural networks [47, 57, 7, 59, 55, 5, 29]. Automatically searching augmentation policies in a data-driven manner [5, 29] is critical to achieve the state-of-the-art result [23, 44]. Although the methods enhance the robustness of networks significantly, there are potentials to improve the performance with data augmentation in the testing phase. We empirically observe that simple deformations of input images at test time cause significant performance drop even the network is trained with advanced data augmentation. Moreover, we verify that there is still a large room to improve the trained network's performance with the appropriate data transformation at the testing phase.

Test-time augmentation of the input data has often been used to produce more robust prediction results. Given a test input image, it averages the network's predictions over multiple transformations

---

[*]Equal Contribution.
[†]Corresponding Author.

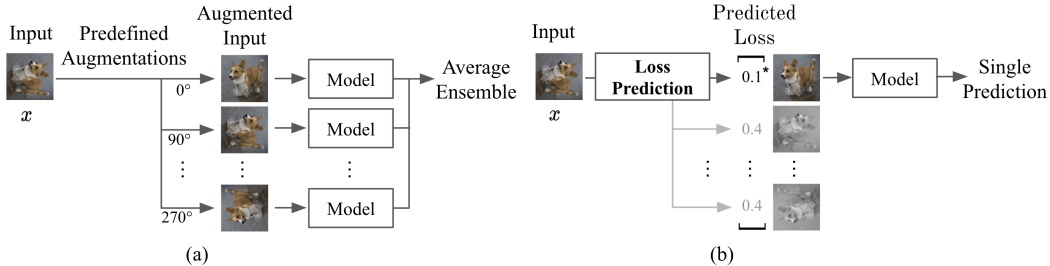

Figure 1: Conceptual comparison between conventional test-time augmentation and the proposed test-time augmentation. (a) Conventional test-time augmentation. (b) Our proposed test-time augmentation. Previous test-time augmentations use prefixed transformations regardless of input. On the other hand, our method predicts the loss value for each transformation before choosing one or a few. Note that this figure shows only one augmentation is selected by predicted losses, i.e. $k = 1$.

for ensemble effect [26, 43, 18]. However, previous test-time augmentation methods have adopted simple geometric transformations such as horizontal, vertical flips, and rotations of 90 degrees [4, 53]. To validate the naive transformations, they augment every input image in substantial amounts [24, 50, 49]. The procedures naturally increase the inference cost at test time. More recently, [32] proposed a learnable test-time augmentation method to find static policies from extended search space. Nevertheless, it performs a greedy search on the test set that is not optimal for each input image. It also requires an average ensemble of dozens of augmented inputs to improve the performance.

In this work, we propose an instance-aware test-time augmentation algorithm. With a pre-trained target network, the proposed method aims to select optimal transformations for each test input dynamically. The method requires measuring the expected effects of each candidate transformation to classify the image accurately. We develop a separate network to predict the loss of transformed images (see Figure 1). Note that the loss reflects both the correctness and the certainty of the classification result. To produce the final classification result, we average the target network's average classification outputs over the transformations having lower predicted losses. We compare the proposed test-time augmentation with the previous approaches on two benchmark datasets. The results demonstrate that our proposed method achieves more robust performances on deformed datasets. The proposed method is efficient: 1) the loss prediction network is compact, and 2) the instance-level selection can significantly reduce the number of augmentations for ensembling. To the best of our knowledge, the proposed method is the first instance-aware test-time augmentation method.

Our main contributions can be summarized as follows:

- We propose the instance-aware test-time augmentation algorithm based on the loss predictor. The method enhances image classification performances by dynamically selecting test-time transformations according to the expected losses.

- The proposed loss predictor can efficiently predict relative losses for all candidate transformations. The predictor makes it possible to select appropriate transformations at the instance level without a high computational burden.

- Compared with predefined test-time augmentation methods, we demonstrate the effectiveness of the proposed method on image classification tasks. Especially, we validate the enhanced robustness of the proposed method empirically against deformations at test time.

## 2 Related Works

**Data Augmentation:** Data augmentation is successfully applied to deep learning, ranging from image classification [6] to speech recognition [1, 17]. Generally, designing appropriate data transformation requires substantial domain knowledge in various applications [41, 1, 59]. Combining two samples, searching augmentation policy, and mixing randomly generated augmentations have been proposed to enhance the diversity of augmented images. Mixup [47, 57] combines two samples, where the label of the new sample is calculated by the convex combination of one-hot labels. CutMix [55] cuts the patches and pastes to augment training images. The labels are also mixed proportionally to the area of

the patches. AutoAugment [5, 29] searches the optimal augmentation policy set with reinforcement learning. AugMix [20] mixes randomly generated augmentations and uses a Jensen-Shannon loss to enforce consistency. Although these training data augmentations have shown promising results, there is still room to improve the performance with testing-phase methods. Moreover, the data augmentation in the training phase is insufficient to handle the deformation in testing data [32].

**Test-Time Augmentation:** Researchers have actively investigated data augmentations in the training phase, transforming data before inference has received less attention. The basic test-time augmentation combines multiple inference results using multiple data augmentations at test time to classify one image. For example, [26, 43, 18] ensemble the predicted results of five cropped images where one is for central and others for each corner of the image. Mixup inference [36] mixups the input with other random clean samples. Another strategy is to classify an image by feeding it at multiple resolutions [18, 41]. [32] introduces a greedy policy search to learn a policy for test-time data augmentation based on the predictive performance over a validation set. In medical segmentation, nnUNet [24] flips and rotates the given input image to generate 64 variants. Then, they use the average of those results to predict. Similar approaches have been studied in [50, 49].

**Uncertainty Measure:** Measuring uncertainty plays an important role in decision making with deep neural networks [13]. In the classification and segmentation tasks, the user makes subsequent revisions in the final decision with estimated uncertainty [45, 28, 34, 33]. Some methods have used the loss estimation [12, 40, 54]. The loss of the trained model for each input implies the certainty of the output for the corresponding input. Therefore, by estimating the loss, we can measure the uncertainty directly. [12, 40] have calculated the loss of training data to improve performance. They regard training data with high losses, which implies high uncertainty, as being important for model improvement. Recently, [54] proposed to predict the loss values to measure the uncertainty of unlabeled ones directly. They added a small auxiliary module to the target network and trained it using the margin-ranking loss. We also measure the uncertainty of the available transformation with a loss prediction module; however, we use a separate module to predict relative loss values for test-time augmentation rather than active learning.

**Robustness in Convolutional Neural Network:** A convolutional neural network is vulnerable to simple corruption. This vulnerability has been studied in several works. [9] explains that even after networks are fine-tuned to be robust against Gaussian noise or blur, the networks lag behind human visual capabilities. [14] explains that the generalization performance of fine-tuned networks for specific corruption is poor. [19] proposes corrupted and perturbed ImageNet datasets and a training-time data augmentation method to alleviate the convolutional neural network's fragility. [48] fine-tuned blurred images and found that fine-tuning for one type of blur cannot be generalized to the other types of blur. [8] solves the under-fitting of corrupted data by using corruption-specific experts. [46] demonstrated the robustness of deep models with the use of mixup. The findings of their paper are, Mixup can be used for improving the certainty in predictions by tempering the overconfident predictions on random Gaussian noise perturbations as well as out-of-distribution images to some extent. [19] found that more representations, more redundancy, and more capacity significantly enhance the robustness of networks against corrupted inputs. Increasing the capacity of networks is simple and effective but requires a substantial amount of computational resources.

## 3 Method

In this section, we describe the proposed method in detail. We begin with the description of the overall test-time augmentation procedure in Section 3.1. Then, we introduce our discretized transformation space in Section 3.2. Our novel loss prediction module and the method to train this module are introduced in Section 3.3 and 3.4, respectively. Details on implementation are in Appendix A.3.

### 3.1 Test-Time Augmentation

We start this section with a formal definition of test-time augmentation. Let $x$ be a given input image and $\tau$ be a transformation operation. If one chooses $\mathcal{T} = \{\tau_1, \tau_2, ..., \tau_{|\mathcal{T}|}\}$ as a candidate set of augmentations at test time, applying conventional test-time augmentation can be formulated as:

$$y_{tta} = \frac{1}{|\mathcal{T}|} \sum_{i=1}^{|\mathcal{T}|} \Theta_{target}(\tau_i(x)),$$ (1)

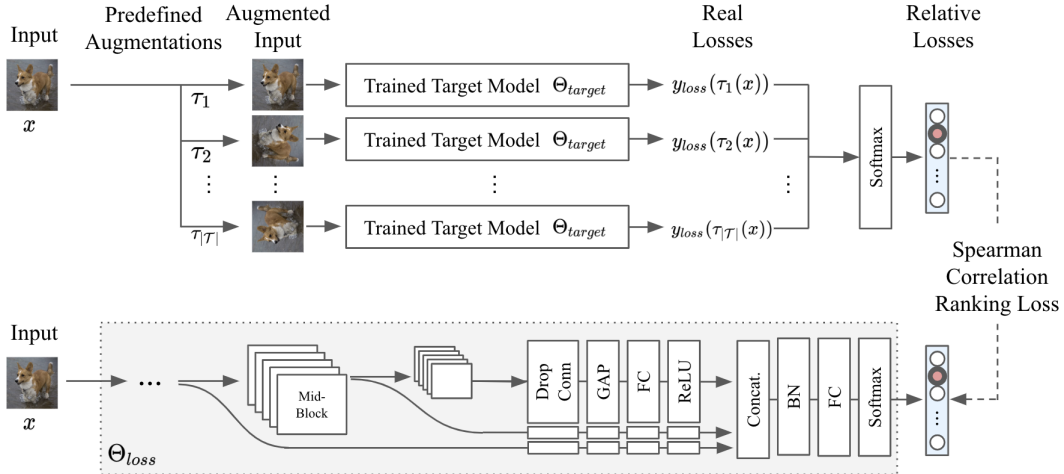

Figure 2: The training process of the proposed loss predictor. The upper part shows how we can get the relative loss values. The lower illustrates our loss prediction network. Given an input $x$, every possible input transformed by candidate transformations is evaluated on the trained target model, $\Theta_{target}$, to get the real loss value. We normalize the real loss values from every transformed $x$ to calculate the relative loss values, $\bar{y}_{loss}$. Our loss prediction network $\Theta_{loss}$ aggregates multi-level features to predict the relative loss values. The loss predictor is trained by ranking loss. Note that we only train $\Theta_{loss}$ while $\Theta_{target}$ is fixed.

where $\Theta_{target}$ is the neural network trained on the target dataset. For example, [18, 26] ensemble 5 different crops and their horizontally flipped versions, which means ensembling by averaging prediction results of 10 images. If one can sort out an optimal set of transformations $\mathcal{T}_k^\star$ from $k$-sized transformation subsets $\mathcal{T}_k = \{\tau_1, \tau_2, ..., \tau_k\} \subseteq \mathcal{T}$, test-time augmentation can be efficiently conducted as:

$$y_{tta}^\star = \frac{1}{|\mathcal{T}_k^\star|} \sum_{i=1}^{|\mathcal{T}_k^\star|} \Theta_{target}(\tau_i^\star(x)), \tag{2}$$

where the total computational cost will be reduced by $\frac{|\mathcal{T}_k^\star|}{|\mathcal{T}|} = \frac{k}{|\mathcal{T}|}$, if the computational cost of determining $\mathcal{T}_k^\star$ can be ignored. As an ideal case, if one wants to improve the performance with the smallest computation, only the best transformation $\mathcal{T}_1^\star$ can be used. The best transformation is expected to enhance the performance of the target network optimally.

Here, we would like to dynamically compose $\mathcal{T}_k^\star$, by selecting top-$k$ transformations having the $k$ lowest predicted loss values given each input image, as shown in Figure 1. To implement the policy, we require an appropriate loss prediction module, $\Theta_{loss}$. In the following subsections, we introduce the loss prediction module to estimate relative loss value for each discretized transformation candidate.

### 3.2 Test-Time Augmentation Space

Diversifying candidate transformations increases the training cost of a loss prediction module because it has to compute ground-truth loss values for every possible transform. In contrast, restricting the diversity of augmentations degrades the effect of data augmentation. Therefore, we design a moderately diverse augmentation space. We adopt two geometric transformations (rotation, zooming), two color transformation (color, contrast), and one image filter (sharpness). We discretize the space within a moderate range: $\{-20°, 20°\}$ rotations; $\{0.8, 1.2\}$ zoomings; color-enhancements; an auto-contrast; and $\{0.2, 0.5, 3.0, 4.0\}$ sharpness-enhancements. To simplify further, we use the union set of those with no transformation as $\mathcal{T}$ and apply the transformations individually, rather than using their combinations i.e $\mathcal{T} = \mathcal{T}_{Rotate} \cup \mathcal{T}_{Zoom} \cup ... \cup \mathcal{T}_{Identity}$, and $|\mathcal{T}| = 12$. Refer to Appendix A.2 for specific parameters and operation methods.

### 3.3 Loss Prediction Module: Architecture

The loss prediction module is an essential part of our method to select promising sets of augmentations. In this subsection, we describe how we design the module to estimate loss values corresponding to each transformation.

One may want to estimate the loss values directly using the target network. Like the previous loss predictors [54], the target network can also predict the loss value as an auxiliary output. However, this requires at least one additional inference on the target network to decide test-time augmentation. It means that a prediction is at least twice as computationally expensive as a prediction without test-time augmentation. Therefore, to enhance efficiency, we use a small neural network completely-separated from the target network. As shown in Figure 2, our loss predictor $\Theta_{loss}$ takes $x$, original input without transformations, and predicts relative losses of each transformed input within the given transformation space $\mathcal{T}$ such that

$$\hat{y}_{loss} = \Theta_{loss}(x), \tag{3}$$

where $\hat{y}_{loss}$ is a $|\mathcal{T}|$-sized vector, representing relative loss values of each transformation $\tau \in \mathcal{T}$. Any neural network can be chosen for $\Theta_{loss}$. We use EfficientNet-B0 [44] since it is the recently proposed state-of-the-art architecture in image classification with much less computations required. We also modify it to utilize multi-level features as in [54]. We expect that the network learns a variety of low-level representations rather than high-level contents only. Note that each feature from the multiple levels can be dropped during training for regularization.

### 3.4 Loss Prediction Module: Training Method

In this subsection, we describe how to train the loss prediction module. Let's say we have a training dataset $D_{train}$, a validation dataset $D_{valid}$, and a fully-trained target network $\Theta_{target}$. The weights of the target network are trained on $D_{train}$ and its performance is evaluated on $D_{valid}$ as an ordinary learning scheme. We freeze the target network $\Theta_{target}$ and split $D_{train}$ into two folds, $D_{loss-train}$ and $D_{loss-valid}$. The first fold will be used to train the loss prediction module and another one will be used to validate the performance. It could be possible to use disjoined training datasets for $\Theta_{target}$ and $\Theta_{loss}$, resulting in reduced training data for the target network. We confirmed that the performance improved marginally in this way, but to increase the usability of our method, we split training data only for the loss predictor.

To generate the output of the loss prediction module, $\hat{y}_{loss}$, we gather the ground-truth loss values $y_{loss}$ for all possible transformations through inferences on the target network. Namely, as shown in Figure 2, the input image is transformed by each of augmentations $\tau \in \mathcal{T}$. Then, the transformed images are fed into the target network to obtain the ground-truth loss value corresponding to $\tau$. Since in the proposed test-time augmentation, only relative losses are necessary, we normalize the loss values by taking the softmax function on both the ground-truth loss values and the predicted loss values. Here, we denote the ground-truth relative loss values as $\bar{y}_{loss}$, and we use the ranking loss surrogates proposed in [10] which directly optimize Spearman correlation between the relative losses as shown in the rightmost part of the Figure 2. Specifically, to optimize the non-differentiable Spearman correlation between relative losses and predictions, we trained a recurrent neural network that approximates the correlation using the official implementation[3] for [10]. We observe that adopting these relative losses with the ranking loss leads to more stable training of the loss prediction network than the exact loss values.

## 4 Experiments

In Section 4.1 and 4.2, we evaluate our method on two classification tasks. We choose CIFAR-100 [25] and ImageNet [6] as standard classification benchmarks and use their corrupted variants, CIFAR-100-C and ImageNet-C [19]. Those corrupted variants consist of 15 types of algorithmically generated corruptions $c$ from noise, blur, weather, and digital categories. They also provide four additional corruptions. Each corruption $c$ has five severity levels, $s \in \{1, 2, 3, 4, 5\}$. We train the

loss prediction module for augmented training images $D_{loss-train}$ with 15 corruptions, and then validate the model on the four additional corruptions. Note that we exclude operations in training-time augmentation which overlap with the corruptions. Top-1 error rate on corruption $c$ with severity $s$ is denoted as $E_{c,s}$.

## 4.1 CIFAR-100 Classification

CIFAR-100 benchmark [25] is one of the most extensively studied classification task, which consists of 60,000 images. All images in the dataset are 32 by 32 pixels in size, and most images are in good quality in that the size of the target object is uniform and centered. As proposed in [19, 20], we use the unnormalized average corruption error, $CE_c = \frac{1}{5} \sum_{s=1}^{5} E_{c,s}$, as the metric.

We set up an experiment to compare our proposed method with the conventional simple test-time augmentations. In Table 1, the method we proposed and the conventional ones were compared. Center-Crop, Horizontal-Flip and 5-Crops are widely-used test-time augmentations [26, 18]. See Appendix A.1 for more details. Random baseline chooses random test-time augmentation for each input within our augmentation space. As the result shows, the random baseline does not improve performance, indicating that our augmentation space is sufficiently diverse. Even though ensemble methods with Horizontal-Flip or 5-Crops require more than twice the computation cost, those methods improve the performance marginally. On the other hand, when the proposed method is used, the performance and the robustness for corrupted data are consistently improved with a negligible computational tax. For example, Wide-ResNet-40-2 [56] trained with Fast AutoAugment [29] scored 45.15% average error rate on 15 corruptions. With the proposed method with $k = 1$, the error rate is lowered to 41.92%. We remark that this requires only 1% more computations for an inference, which is quite efficient compared to the conventional methods or the recently-proposed one [32]. Our method can also leverage the ensemble technique, which can achieve 39.90% at a cost about four times that of the Center-Crop. The conventional 5-Crop ensemble costs more, but the error rate is 45.27%.

When learning the loss predictor in the above setting, we can consider that some information about corruption was given, so we added the corruption to training-time augmentation for a comparison. We add 15 corruption operations to training-time augmentations of AugMix, which denotes AugMix+. Augmentation operations, including corruptions, are uniformly-sampled for AugMix, in the same way as other augmentations. The experiment result is shown in Table 2. As expected, performance has improved significantly in most corruptions. However, for held-out corruptions, which are excluded in training, performance improvements have not been generalized, and most importantly, performance fell noticeably for the clean dataset. In the case of our method training loss predictors with 15 corruptions, it is superior to AugMix+ in terms of stability. The proposed method has better generalization on the held-out corruption and no degradation on the clean test-set.

As a particular case, we designed an experiment that assumed prior knowledge of a specific type of corruption, such as blur. We train AugMix and Fast AutoAugment models with four blur corruptions as an additional training-time augmentation case. We also use the same corruption when we train our loss predictor. In Table 3, our method improves the robustness on blur corruptions by a large margin. Remarkably, the proposed method maintains the performance of the clean test-set, regardless of the training-time augmentation method. On the other hand, when blur corruptions are used as training-time augmentation without our test-time augmentation, the performance degradation on the clean test-set could not be prevented.

## 4.2 ImageNet Classification

ILSVRC 2012 classification benchmark (ImageNet) [6] consists of 1.2 million natural images of 1000 classes. We resize the input image for loss predictor $\Theta_{loss}$ to 64 by 64 pixels to reduce computational overhead. Other hyperparameters and training settings are same as all experiments as mentioned in Section 4.1 and A.3. Our loss predictor $\Theta_{loss}$ requires negligible computations for inference compared to the target networks, so the relative cost is almost proportional to the number of ensembles.

Table 4 shows performances with the baselines and the proposed test-time augmentation on ResNet-50 [18]. Our method outperforms by a clear margin over baseline methods. For models trained with AugMix, ensemble of two augmented images by our methods (i.e. $k = 2$) lowers $mCE$ for

Table 1: Evaluation result on CIFAR-100(-C) dataset. Metric for corrupted set is average corruption error, $mCE = \frac{1}{|c|}\sum_c CE_c$.

| Model | Train-time Augmentation | Test-time Augmentation | Relative Cost | Clean Test-set | Corrupted set | Corrupted test-set |
|---|---|---|---|---|---|---|
| Wide-ResNet [56] | AugMix [20] | Center-Crop | 1.00 | 23.34 | 36.15 | 33.19 |
| | | Horizontal-Flip | 2.00 | 22.86 | 35.10 | 32.17 |
| | | 5-Crops | 5.00 | 22.66 | 35.80 | 32.66 |
| | | Random ($k=1$) | 1.00 | 28.57 | 41.59 | 39.50 |
| | | Random ($k=2$) | 2.00 | 25.45 | 38.06 | 35.85 |
| | | Random ($k=4$) | 4.00 | 23.84 | 35.94 | 33.62 |
| | | Ours ($k=1$) | 1.01 | 23.31 | 33.15 | 30.84 |
| | | Ours ($k=2$) | 2.01 | 23.19 | 33.03 | 30.80 |
| | | Ours ($k=2$) + Flip | 4.02 | 22.69 | 32.76 | 30.11 |
| | Fast AutoAug [29] | Center-Crop | 1.00 | 21.39 | 45.15 | 41.59 |
| | | Horizontal-Flip | 2.00 | 20.84 | 44.28 | 40.71 |
| | | 5-Crops | 5.00 | 21.46 | 45.27 | 41.63 |
| | | Random ($k=1$) | 1.00 | 27.30 | 47.15 | 43.49 |
| | | Random ($k=2$) | 2.00 | 25.55 | 48.22 | 45.57 |
| | | Random ($k=4$) | 4.00 | 23.33 | 45.60 | 43.06 |
| | | Ours ($k=1$) | 1.01 | 20.47 | 41.92 | 37.57 |
| | | Ours ($k=2$) | 2.01 | 20.43 | 40.84 | 36.78 |
| | | Ours ($k=2$) + Flip | 4.02 | 20.41 | 39.90 | 35.89 |
| ResNext [52] | AugMix [20] | Center-Crop | 1.00 | 20.87 | 33.73 | 31.46 |
| | | Horizontal-Flip | 2.00 | 20.17 | 33.20 | 33.06 |
| | | 5-Crops | 5.00 | 20.64 | 33.52 | 31.09 |
| | | Random ($k=1$) | 1.00 | 25.37 | 39.03 | 35.66 |
| | | Random ($k=2$) | 2.00 | 22.83 | 35.61 | 34.04 |
| | | Random ($k=4$) | 4.00 | 21.05 | 33.67 | 31.98 |
| | | Ours ($k=1$) | 1.00 | 20.94 | 32.32 | 30.29 |
| | | Ours ($k=2$) | 2.00 | 20.91 | 32.42 | 30.35 |
| | | Ours ($k=2$) + Flip | 4.00 | 20.22 | 31.90 | 29.78 |

Table 2: Comparison with training-time augmentation when corruptions are expected in the test phase. Adding severely corrupted images to training data by training-time data augmentation can affect adversely. AugMix+ is trained with corrupted images in training time. All metrics are top-1 error rates (for corrupted test sets, we average for 5-severity levels). It is bold when there is an improvement of 1% or more, and red when there is more than 1% degradation in performance.

| | Clean | Noise | | | Blur | | | | Weather | | | Digital | | | | | Held-out Corruptions | | | |
|---|---|---|---|---|---|---|---|---|---|---|---|---|---|---|---|---|---|---|---|---|
| Method | Test-set | Gauss.n | Shot | Impul. | Defoc. | Glass | Motion | Zoom | Snow | Frost | Fog | Bright | Contr. | Elastic | Pixelate | JPEG | Speckle | Gauss.b | Spatter | Satur. |
| AugMix | 23.34 | 54.32 | 45.76 | 38.10 | 25.56 | 49.60 | 28.93 | 28.06 | 34.09 | 37.18 | 33.15 | 26.35 | 34.42 | 32.67 | 33.57 | 39.75 | 43.07 | 26.85 | 27.67 | 35.17 |
| AugMix+ | 25.16 | **31.46** | **29.71** | **27.28** | 27.06 | **35.33** | 28.93 | 27.09 | **30.34** | **29.03** | **31.57** | 27.35 | **30.87** | 31.89 | **29.56** | **33.66** | **29.56** | 27.52 | 29.33 | 35.54 |
| Ours($k=1$) | 23.31 | **45.76** | **29.77** | **36.78** | 24.93 | **42.42** | 27.89 | 26.85 | **34.71** | 36.45 | 33.23 | 26.37 | **30.78** | 32.70 | **28.93** | 39.75 | **38.68** | 25.82 | 25.99 | 32.88 |

(known-)corruptions from $67.72\%$ to $64.55\%$. Also, it shows generalization performance similar to the conventional 5-Crops at less than half the computational cost.

We also compare the proposed method with the recently proposed test-time augmentation method, GPS [32] using their official code[4]. Moreover, we search policies directly on the corrupted dataset using GPS to see how well it responds to corruption and name it GPS†. Experimental results show that the proposed method outperforms both GPS and GPS† on ImageNet-C. This means that the performances of GPS on both seen and unseen corruptions lag behind our proposed method. In particular, the GPS policies found on the corrupted dataset produce poor results in the clean set, while our method prevents the performance degradation on the clean set. We confirm by the GPS code that the search space of GPS includes all our augmentation policies such as "auto-contrast"

Table 3: Evaluation after trying to improve performance against blur corruptions. It is bold when there is an improvement of 1% or more, and red when there is more than 1% degradation in performance.

| Method | Clean Test-set | Noise | Blur | Weather | Digital | Avg. | Speckle | Gauss.b | Spatter | Saturate |
|---|---|---|---|---|---|---|---|---|---|---|
| | | Corruptions | | | | | Held-out Corruptions | | | |
| Fast AutoAug | 21.39 | 53.82 | 44.02 | 35.66 | 49.26 | 45.15 | 55.58 | 47.86 | 27.14 | 35.76 |
| Fast AutoAug (\w Blur) | 22.41 | **49.69** | **26.48** | **33.76** | **40.89** | **36.90** | **49.51** | **24.48** | 27.91 | 36.44 |
| AugMix | 23.34 | 46.06 | 33.04 | 32.89 | 35.10 | 36.15 | 43.07 | 26.85 | 27.67 | 35.17 |
| AugMix (\w Blur) | 25.08 | 58.62 | **38.69** | 36.54 | 42.38 | 43.09 | 53.27 | 26.12 | 36.90 | 38.22 |
| Fast AutoAug + Ours | **21.44** | 53.77 | **34.85** | 35.71 | **47.33** | 42.20 | 55.56 | **31.58** | 27.31 | 35.89 |
| AugMix + Ours | 23.35 | **44.94** | **30.49** | 32.83 | 34.17 | **34.98** | 42.37 | 25.98 | 27.74 | 35.18 |

Table 4: ImageNet dataset evaluation result on ResNet-50. GPS†: Greedy Policy Search on the corrupted dataset.

| Train-time Augmentation | Test-time Augmentation | Relative Cost | Clean Test-set | Corrupted set mCE | Corrupted Test-set mCE |
|---|---|---|---|---|---|
| Standard | Center-Crop | 1 | 24.14 | 78.93 | 75.42 |
| | Horizontal-Flip | 2 | 23.76 | 77.91 | 74.32 |
| | 5-Crops | 5 | 23.91 | 77.52 | 73.87 |
| | 10-Crops | 10 | 23.04 | 76.69 | 72.98 |
| | Random($k{=}1$) | 1 | 26.89 | 82.86 | 79.81 |
| | Random($k{=}2$) | 2 | 25.14 | 79.91 | 77.00 |
| | Random($k{=}4$) | 4 | 24.29 | 78.24 | 75.38 |
| | GPS($k{=}1$) | 1 | 24.86 | 82.13 | 79.43 |
| | GPS($k{=}2$) | 2 | 23.78 | 76.45 | 73.32 |
| | GPS($k{=}4$) | 4 | 23.44 | 77.27 | 73.87 |
| | GPS†($k{=}1$) | 1 | 27.39 | 77.21 | 75.07 |
| | GPS†($k{=}2$) | 2 | 27.04 | 76.48 | 74.27 |
| | GPS†($k{=}4$) | 4 | 26.88 | 76.09 | 73.84 |
| | Ours($k{=}1$) | 1 | 24.14 | 75.52 | 74.29 |
| | Ours($k{=}2$) | 2 | 24.10 | 75.00 | 73.61 |
| | Ours($k{=}2$) + Flip | 4 | 23.74 | 74.00 | 72.59 |
| AugMix [20] | Center-Crop | 1 | 22.45 | 67.72 | 64.67 |
| | Horizontal-Flip | 2 | 22.23 | 67.72 | 65.91 |
| | 5-Crops | 5 | 21.71 | 66.19 | 63.32 |
| | 10-Crops | 10 | 21.54 | 65.67 | 62.76 |
| | Random($k{=}1$) | 1 | 24.09 | 71.07 | 72.87 |
| | Random($k{=}2$) | 2 | 23.04 | 68.82 | 66.33 |
| | Random($k{=}4$) | 4 | 22.73 | 67.30 | 64.88 |
| | Ours($k{=}1$) | 1 | 22.38 | 65.02 | 63.92 |
| | Ours($k{=}2$) | 2 | 22.37 | 64.55 | 63.39 |
| | Ours($k{=}2$) + H-Flip | 4 | 22.10 | 63.90 | 62.71 |
| | Ours($k{=}2$) + 5-Crops | 10 | 21.66 | 63.50 | 62.33 |

and "sharpness"; our performance gains come from the proposed instance-specific transformation. As reported, GPS [32] ensembles prefixed 20 test-time augmented images to improve mCE of the ResNet-50 model from 68.7% to 67.3%. Our method requires much less cost to achieve this relative improvement, and a much higher level of performance is also possible. Figure 3 shows selected examples, which are classified correctly after test-time augmentation.

## 5 Discussion

**Possibilities.** In Appendix A.4, we demonstrate that performance can be significantly improved, assuming the lowest loss among augmented images can be selected. In the clean test-set, it beats the existing state-of-the-art models by a clear margin. Also, for the corrupted test-set, such as

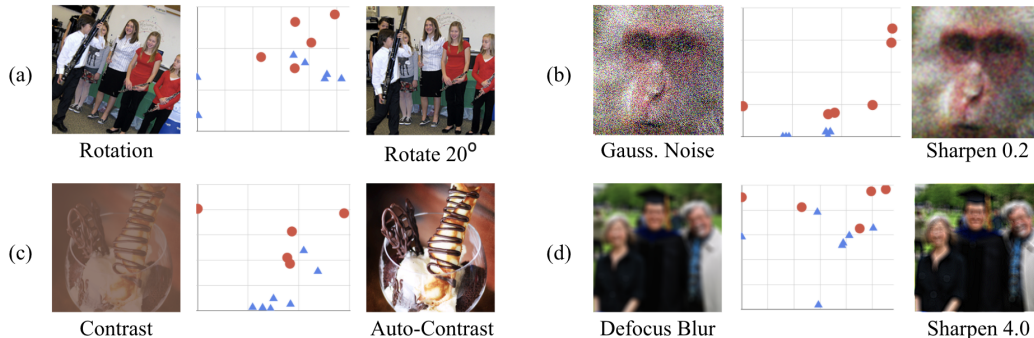

|  | Rotation | Rotate 20° |  | Gauss. Noise | Sharpen 0.2 |
| (a) | | | (b) | | |
| (c) | Contrast | Auto-Contrast | (d) | Defocus Blur | Sharpen 4.0 |

Figure 3: Selected Examples which are correctly classified by the proposed method. Left: Corrupted image, Middle: Scatter plot of predictions (x) and actual loss values (y), Right: Test-time augmented image by our method. In the Scatter Plot, the red circle corresponds to the augmented image that matches the label, and the blue triangle does not. Note that it was processed to exaggerate the corruption than the actual image in the case of (b) and (d).

ImageNet-C and CIFAR-100-C, models can be close to clean data performance. Also, a well-designed augmentation space will be beneficial. In this work, we found some augmentation operations contributed to some specific corruptions, e.g., sharpness enhancements for noise corruptions. By adding sophisticated augmentation operations with a clear purpose, e.g., deblur for blurred images, it is expected that large performance improvement can be achieved.

**Learning Loss Prediction Module.** As the accuracy of the learned loss predictor increases, the performance of the proposed method may increase, so we examined how well the learned loss predictor works. For ImageNet experiments, loss predictions from our loss predictors have an average correlation of 0.38 on validation set $D_{loss-valid}$ and 0.30 on test set $D_{test}$. In [54], the correlation between predicted losses and ground-truth values for unseen data was 0.68, but our loss prediction seems to be a more difficult problem because we predict the change of loss by transformation for the same data. Since our method predicts a loss with a rather weak correlation, it works well for ensembles when $k \leq 2$ .

**Limitations.** One of the things we discovered while conducting various experiments was that it was ineffective to train the loss predictor for some target models. For example, we were unable to train appropriate loss predictors for models trained with CutMix. It was also difficult to train a loss predictor that works well for some corruptions included in training-time augmentation. For those cases, we conjecture that loss values are quite noisy to predict as predictions from a neural network would fluctuate by a small perturbation [19]. Further investigation on these problems could be essential to make instance-aware test-time augmentation remarkably successful.

## 6 Conclusion

We propose a novel instance-aware test-time augmentation. The method introduces a loss prediction network to determine appropriate image transformations in test time. Unlike the previous ensemble-based methods, the proposed method is computationally efficient and practical to use in real-world situations. The loss prediction module enhances the classification accuracy and robustness against deformation in test time with a low computational cost. Although the method shows promising results, there is still room for improving the method. In this paper, we used a limited number of transformations and their specifications. To enhance robustness in real-world situations, we plan to study more general settings for loss prediction networks. Reinforcement learning methods are good candidates to optimize the loss prediction networks efficiently. Since the proposed framework can apply to other domains, we will validate our method in various image tasks such as object detection and segmentation. Moreover, examining the relationship between training- and test-time augmentation is also crucial in future work because sophisticated combining of two augmentations can further improve the performance of neural networks. We also expect future works to be conducted to use less cost while including more augmentations.

## Broader Impact

Regardless of the state of the given data, using it as an input of the neural network with the same pre-processing can be ineffective in terms of performance and stability. We propose a novel instance-aware test-time augmentation. If a deep learning model is in the deployment stage, it can be expected to increase stability and performance through the proposed method. However, it is still necessary to verify the stability and generalization of the proposed method because it is in an early stage of research for instance-aware test-time augmentation. At this time, the performance of deep learning models may be degraded in unexpected situations, although we haven't described specific conditions.

## Footnotes

[3]https://github.com/technicolor-research/sodeep

[4]https://github.com/bayesgroup/gps-augment

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
