[Supplementary Material]

# A    Appendix

## A.1    Conventional Test-Time Augmentation

Center-Crop is the standard test-time augmentation for most of computer vision tasks [56, 29, 5, 7, 18, 26, 52]. The Center-Crop first resizes an image to a fixed size and then crops the central area to make a predefined input size. We resize an image to 256 pixels and crop the central 224 pixels for ResNet-50 in ImageNet experiment, as the same way as [18, 26, 52]. In the case of CIFAR, all images in the dataset are 32 by 32 pixels; we use the original images without any modification at the test time. Horizontal-Flip is an ensemble method using the original image and the horizontally inverted image. [18, 26] used 5/10-Crops for ImageNet. The 5-Crops ensembles 5 images by combining the center crop with crops from 4 corners. 10-Crops is an average ensemble by adding horizontally flipped images to those from 5-Crops. Feeding images in multiple resolutions are used for test-time augmentations in ResNet [18] and VGGNet [41]. However, these methods were ineffective in our experiment. Generally, existing methods aggregate the results of safe transformations, which are included in training-time augmentation.

## A.2    Test-Time Augmentation Space

As we explain in Section 3.2, our test-time augmentation space consists of 12 operations. Figure 4 shows a selected data sample and its augmented versions. We implemented image transformations using a popular Python image library.[5] The rotation and zoom operations are applied with the bicubic resampling option. $\mathcal{T}_{AutoContrast}$ is implemented by using PIL.ImageOps.autocontrast function, which normalizes the image contrast with color histogram. $\mathcal{T}_{Sharpness}$ is to adjust the sharpness of a given image. PIL.ImageEnhance.Sharpness function gives a blurred image with parameters less than 1.0 and a sharpened image with parameters greater than 1.0. We used PIL.ImageEnhance.Color function for $\mathcal{T}_{Color}$ changing image color balance. A transformed image might contain unexpected clues to know what transformations have been applied. For instance, a rotated image will have blank areas implying how much the rotation took place. We mitigate this problem by filling blank areas with mirror-ed contents of the image.

| $\mathcal{T}_{Identity}$ | $\mathcal{T}_{Rotate: -20°}$ | $\mathcal{T}_{Rotate: +20°}$ | $\mathcal{T}_{Zoom: 0.8}$ | $\mathcal{T}_{Zoom: 1.2}$ | $\mathcal{T}_{AutoContrast}$ |
|---|---|---|---|---|---|
| 0.086 | 0.049 | 0.099 | 0.15 | 0.098 | 0.055 |

| $\mathcal{T}_{Sharpness: 0.2}$ | $\mathcal{T}_{Sharpness: 0.5}$ | $\mathcal{T}_{Sharpness: 2.0}$ | $\mathcal{T}_{Sharpness: 4.0}$ | $\mathcal{T}_{Color: 0.5}$ | $\mathcal{T}_{Color: 2.0}$ |
|---|---|---|---|---|---|
| 0.139 | 0.119 | 0.037 | 0.038 | 0.091 | 0.039 |

Figure 4: A selected ImageNet sample from swing class and the corresponding (relative) loss values from fully-trained ResNet-50 after transformations. The original image was distorted by some corruptions, such as rotation and noise. The numbers below each transformed image are relative loss values obtained by inference on the target network and SoftMin normalization. The Blue border indicates that the target network predicts the label correctly. The noisy image can be advantageous in test-time after zoom-out ($\mathcal{T}_{Zoom: 0.8}$) or blur ($\mathcal{T}_{Sharpness: 0.2}$, $\mathcal{T}_{Sharpness: 0.5}$). Moreover, it was helpful to rotate the image ($\mathcal{T}_{Rotate: +20°}$) into the normal orientation. Note that it was processed to exaggerate the blur corruption than the actual image in the case of $\mathcal{T}_{Sharpness: 0.2}$ and $\mathcal{T}_{Sharpness: 0.5}$, so that the difference can be seen.

Table 5: Top-1 Accuracy (%) on CIFAR-100 and ImageNet with random-TTA and oralce-TTA. The performances obtained by oracle-TTA indicate the upper bounds for the proposed TTA with our transformation space, and it shows high potential of performance improvement by our TTA since it is much better than state-of-the-art models without external data.

| Dataset | Model | Fast AutoAugment [29] | With Test-Time Augmentation | |
|---------|-------|----------------------|-----------------------------|---|
| | | | Random-TTA | Oracle-TTA |
| CIFAR-100 | Wide-ResNet-40-2 | 79.3 | 73.7 | 90.7 |
| | PyramidNet-272 | 88.1 | 82.0 | 92.8 |
| ImageNet | ResNet-50 | 77.6 | 73.1 | 87.6 |

## A.3 Implementation

Our neural network implementation depends on PyTorch [37]. Gathering the ground-truth loss values of all possible transformations requires applying each of the transformations and performing the respective inference on the target network. For this, intensive computational resources are essential to prepare training data. Thus, we parallelize DataLoader in PyTorch by customizing it to perform image transformations and loss evaluations on distributed computing nodes[6]. This parallelization greatly reduces the learning time, even though it does not reduce the total computational cost.

When training the loss prediction module, we set the hyper-parameters as same as possible: learning rate as 0.01 and cosine annealing [30]; batch size as 512; Dropout [42] ratio 0.3 and Drop-Connect [22] ratio 0.2 for better generalization; RMSProp optimizer with decay 0.9 and momentum 0.9; batch norm momentum 0.99; weight decay 1e-5; exponential moving average of weights with momentum 0.999. We use up-to 64 nodes to parallelize data-generating process. We use the official implementation code[7] for [10].

Besides, we perturb training data itself to avoid overfitting. Similar to [21], we augment the same data multiple times in a single batch. In the same batch, we have images with the same content but requiring different test-time augmentation. This helps to avoid overfitting by guiding to focus on the image state rather than the content of the image. Applying Cutout [7] to up to half of the image also helps.

## A.4 Test-Time Augmentation Oracle

To examine the proposed method's maximum performance improvement, an instance-aware test-time augmentation, we suggest a hypothetical loss predictor called Oracle-TTA. It is assumed that we can know the relative loss accurately so that testing images are augmented by the transformation with the lowest loss. On the other hand, if the loss predictor performs inadequately, there is no difference between random selection and the proposed method. We call this random selection as Random-TTA.

As Table 5 shows, the two hypothetical loss predictors show significantly different performance. Random-TTA reduces performances as expected. In contrast, Oracle-TTA enhances performances in a clean test-set with a considerable margin. The tendency is retained for corrupted datasets. The result implies that it is possible to achieve exceptionally high performance using instance-aware test-time augmentation. To the best of our knowledge, our proposed method is the first work in instance-aware test-time augmentation. We are particularly looking for possibilities in subsequent studies that can optimize training-time augmentation and test-time augmentation jointly.

## A.5 Analysis of Selected Test-Time Augmentation for ResNet-50 on ImageNet

In order to see which test-time augmentation has contributed to performance improvement, the operation's selection ratio is plotted in Figure 5. In the clean test-set, $\mathcal{T}_{Identity}$ is selected for 75% of test images. There was a slight performance improvement for the rest of the test images by applying contextual test-time augmentation, such as AutoContrast. For Gaussian noise corruption, blurring ($\mathcal{T}_{Sharpness:0.5}$) or zoom-out ($\mathcal{T}_{Zoom:0.8}$) operation is selected when the severity is relatively low.

On the other hand, if the severity is high, the stronger blur effect ($\mathcal{T}_{Sharpness:0.2}$) is used to drive performance improvement. For defocus blur corruption, sharpen effect ($\mathcal{T}_{Sharpness:3.0}$) or zoom-in ($\mathcal{T}_{Zoom:1.2}$) operation is chosen, which makes sense intuitively. Furthermore, as severity increases, the rate at which these operations are selected also increases. As in these examples, you can see that the test-time augmentation is determined according to the type of corruption or its severity. This is an instance-aware test-time augmentation that is different from conventional methods. It is also generalized to new types of corruption that have not been seen in the learning of the loss predictor.

Figure 5: Selection rate of test-time augmentation by corruption and severity. It shows that the test-time augmentation operation that contributes to the performance varies depending on the corruption and severity. This is the difference that the proposed method, a novel instance-aware test-time augmentation, has compared to the existing methods.

## A.6 Test-time Augmentation for The Clean Set

We conducted experiments with our loss predictor trained for the clean set. In Figure 6, our loss predictor picks out promising one out of five crop regions. Even if only one crop region is selected using the loss predictor, the obtained performance is comparable to the existing 5-crop ensemble. This is clear proof that our method is also effective on the clean set and separated from our search space; our loss predictor itself can contribute to enhancing the classification performance. In Figure 7, instead of using 5-Crop augmentations, we trained our loss predictor on the optimized GPS policies. GPS applies a predefined policy set regardless of the input, but by applying our method additionally, you can select a policy that fits the input from among them; this suggests that there is a possibility of research to improve performance even in a clean dataset.

Figure 6: Comparison for the same 5-Crop candidates on the clean ImageNet set using ResNet-50. Top-1 accuracies by the number of ensembles. We trained our loss predictor for five crop areas. Compared to the 5-crop ensemble, choosing one transform by our method gives almost the same performance, and selecting the two transforms achieves even better performance with less computational cost.

Figure 7: Comparison for the same GPS transforms on the clean ImageNet set using ResNet-50. Top-1 accuracies by the number of ensembles. We trained our loss predictor on the searched GPS policies to choose ones specific for each test instance. Our method properly selects valid transforms from the candidates chosen greedily by GPS, and therefore further improves the performances over static ensemble from GPS.

## Footnotes

[5]https://pillow.readthedocs.io

[6]https://github.com/ildoonet/remote-dataloader

[7]https://github.com/technicolor-research/sodeep