[Reviews · NeurIPS 2020]

Review 1

Summary and Contributions: The paper proposed a new instance-aware (kind of an image adaptive) test-time augmentation technique. The technique uses a pre-trained target network to dynamically select appropriate transformations or each input test image. The pre-trained network calculates the loss of each transformation. Final predictions are the averages of all the images with lowest losses. The empirical studies of the paper showed that the classification performance (reduced test error) was improved.

Strengths: Strengths: 1. Overall, the paper is well written and easy to follow. 2. The new idea of instance-aware test-time augmentation is interesting and has some practical use, especially at places where privacy is of utmost priority, where the inference data is not available. 3. Classification errors decreased significantly compared to the other methods in the results.

Weaknesses: 1. In the Robustness of deep models, the recent NeurIPS 2019 paper Thulasidasan, S.; Chennupati, G.; Bilmes, JA; Bhattacharya, T.; Michalak, S.; On mixup training: Improved calibration and predictive uncertainty for deep neural networks, Advances in Neural Information Processing Systems, 13888-13899, 2019 demonstrated the robustness of deep models with the use of mixup. The findings of their paper are, Mixup can be used for improving the certainty in predictions. Temper the overconfident predictions on random gaussian noise perturbations as well as out-of-distribution images to some extent. Which is missing in the proposed paper. 2. Loss is never defined, please provide an equation or some explanation for better understanding. Is it simply cross entropy loss? It is confusing now because no information is available. 3. For the robustness, the improvement in the test error is looking good, however, what happens to the confidence of predictions? Can we actually say that the test predictions are made with high confidence and high accuracy (low test error, in this case) compared to the baselines? Having some kind of expected calibration error measures is a useful insight. In fact [30] used the calibrated log-likelihood from Ashukha, Arsenii, Lyzhov, Alexander, Molchanov,Dmitry, and Vetrov, Dmitry. Pitfalls of in-domain uncertainty estimation and ensembling in deep learning.In ICLR, 2020. 4. I might be missing something here, how does the current method perform compared to the existing test-time augmentation method in [30]. If some of the experiments in the current version of the paper point out to that, please explicitly specify that otherwise, it is encouraged to compare against the results of [30], having such comparisons will help the paper. Provided the above concerns are addressed, I can change the score.

Correctness: The claims, method and the empirical evaluations appear correct with enough details on the experimental validations.

Clarity: Overall, the paper is well written and easy to follow.

Relation to Prior Work: Yes, the paper clearly distinguishes the contributions with the existing work. "The proposed method is the first instance-aware test-time augmentation method." The paper differentiates the fact that the proposed method is adaptive to select near optimal transformations compared to the existing methods that select all the available transformations. However, one can argue that the greedy policy based approaches in a way are also instance aware, since those methods also explore through the space of transformations. In any case, the paper clearly has distinctions with the literature.

Reproducibility: Yes

Additional Feedback: Although the paper argues that the cost of test-time augmentations are relatively cheap compared to the existing techniques (such as the greedy policy based search heuristics in the space of transformations), there is still room for improvement in reducing the cost and improving the classification performance.


Review 2

Summary and Contributions: The paper proposes a method that improves on test-time augmentation by using a small network that predicts which augmentations provide the best prediction for each image instance. Then this augmentations are used at test-time reducing the inference cost. This approach improves test accuracy in vision datasets (CIFAR100 and ImageNet) in both clean and corrupted scenarios.

Strengths: The idea is novel and has the potential to reduce the inference cost of test-time augmentation by selecting only few promising augmentations (1 or 2) instead of brute-force (>10 augmentations). The authors performed a complete literature review and position well the paper. The authors show strong results in challenging datasets and test several architectures. The authors provide a small section on the limitations of their approach.

Weaknesses: The main weakness of the paper is not being able to separate the effect of "their choice of data augmentations" and "their method of selecting the k-best augmentations" in explaining the performance gain observed in the clean/corrupted datasets. Since they propose the latter, it would be nice to separate the effects and focus on the method. This could be done by the following: Not introducing new augmentations. They can use 5-crop x 2 right-left flips (=10) and use their method to predict which of the 10 augmentations produce the lowest loss. This can already improve the efficiency with respect to brute-force approach of using 10 augmentations without introducing new distortions. This may be challenging, as the authors note in the limitations section, since the network will learn to be invariant over these transformations in the training set (as crops and flips are used at training time). The authors failed to study their data augmentations. What is the performance of using them at training time? If they are used at training time, it is still necessary to have the loss-prediction network or all the gain is explained by the introduction of new augmentations? Finally, the choice of data augmentations can be particularly chosen to "undo" the corruptions of the corrupted dataset and artificially boost the performance in the corruption dataset. Their chosen augmentations include "auto-contrast"and "sharpness" that can potentially revert the distortions "contrast"and "Gaussian noise"present in the corruption test set. Again, it is hard to separate the gains from the prediction network and the data augmentations introduced in section 3.2. One of the fair comparisons is the random baseline, where they show that their method is able to pick a better prediction than the one coming from a random augmentation. I suppose this can be an easy task if one of the augmentations degrade a lot performance. ==UPDATE== The authors addressed the weaknesses I pointed out in my initial review. 1) Although the improvement is small compared to their chosen augmentations, their method also improves accuracy when using training time augmentations which is very attractive to reduce the inference cost of test-time augmentation. 2) I believe the robustness improvements in imagenet-c and cifar-10c benefit from the author's particular choice of augmentations which revert some of the corruptions. At first it seems like a weakness, but the authors pointed that just using the chosen augmentations do not lead to improvement and their prediction network is necessary to pick the best augmentation for each particular instance..

Correctness: The empirical claims are not convincing. I cannot separate the effect of the "prediction network"and "the test augmentation space". ==UPDATE== The authors addressed my concerns,

Clarity: The paper structure makes it easy to follow and read.

Relation to Prior Work: Yes.

Reproducibility: Yes

Additional Feedback: I think the paper proposes an interesting novel idea, however I would recommend to address the weaknesses so the readers can be confident on the value of the proposed approach. A proper ablation of the prediction network effect is necessary. The idea of using a pre-trained network and improving the performance without retraining by just adding a small prediction network and some set of augmentations may have applications elsewhere. The focus of the paper in the corruptions CIFAR-100C and Imagenet-C is in my opinion is unfortunate because the choice of test augmentations that can revert the original corruptions makes it difficult to separate where the performance gains come from. ==UPDATE== The authors included the ablation in their rebutal.


Review 3

Summary and Contributions: The Authors propose a novel test-time augmentation scheme, based on a loss-prediction network that predicts the loss a model incurs when the sample is processed with different transformations. This network can be used to pick the test-time augmentation that causes the lowest expected loss. The proposed method is empirically proven to perform better than standard baselines such as ensembling over random crops/flips in a variety of benchmarks.

Strengths: Test-time augmentation is a largely unexplored topic, and the Authors propose a well-thought approach to face it. Results are encouraging, showing that models endowed with the devised approach are more robust against standard perturbations.

Weaknesses: The main weakness of this work is that the proposed method is not compared against the main competitor (Malchausen et al. [30]). Such method is very related, and the claim made by the Authors in line 36 is not accurate ("it performs a greedy search on the test set that is not optimal for each input image"): the method in [30] finds a policy via greedy search on a subset of validation samples, that is then kept fixed at test phase (see Section 4 in the original paper, "Policy search" paragraph). It is true that it is more computationally demanding at test-time - since the method proposed by the Authors in the manuscript works well even with a single test-time transformation - but the performance of the two methods should be compared. Concerning the results, there are a few experiments that would be very valuable to more thoroughly assess the performance of the method. (i) A "Random" baseline with larger values of $k$ - for example, k=5; is the proposed method still significantly better performing, when the number of test-time augmentations averaged is increased? (ii) Results associated with non-corrupted samples (clean test set). On a minor note, (iii) it would be nice to see results related to models that were not trained with sophisticated augmentation strategies such as AutoAugment etc.

Correctness: The only flaw I could detect is the one reported in the Weaknesses section (point 1).

Clarity: The exposition is clear, but the writing style could be overall improved.

Relation to Prior Work: Yes.

Reproducibility: Yes

Additional Feedback: I would like to add that, despite the weaknesses I spotted led to the a rating of 4/10, I believe that this paper has potential. With more experiments, proper comparisons, and some polishing, this could be a strong submission. Below, some more comments. ----------------- Figure 2, top: the output lines merging into a single one is confusing, because in practice the outputs are processed separately. I would suggesting keeping them separated. Line 182&192: what do the Authors mean with "we train loss predictors using 15 corruptions"? It should be trained with the 12 test-time image transformations, correct? Line 73: this is a subjective statement (namely, missing experiments or supporting references). Line 143-146: this part is not clear to me. Line 152: "state-of-the-art" in which task? Line 164: what does it mean "similar to the common practice in meta-learning"? Line 256: I always appreciate Authors discussing the limitations of their approach. Yet, a claim such as "it was ineffective to train the loss predictor for some target models" should be expanded; in which conditions did it not work? Typos: Line 14: missing verb after "unless large amounts ..." Line 22: "are" -> "is" Line 25: "even the network" -> "even is the network" Line 221: missing verb after "prevents the performance" ("to drop"?) ---------------------------------------- EDIT The Author response addressed some of my concerns. After the discussion with the other Reviewers, I am happy to raise my score to 6.


Review 4

Summary and Contributions: The Authors propose a novel technique to do test-time augmentation which trains a light weight module to estimate loss values over a set of pre-fixed transformations. Inference with transformations with the lower estimated losses leads to improved classification. The authors validate their method on CIFAR-100C and Imagenet-C and show improvements.

Strengths: Improvements are observed when using the proposed method on CIFAR-100C and Imagenet-C over Fast AutoAug and AugMix. Authors show the efficacy of their methods on numerous corruptions (used for images). The simplicity of the method enables the usage of the proposed technique in a plethora of tasks.

Weaknesses: As the authors note, the method does not show promise on corruptions that are included during train-time. This limits the usage of the proposed method. The proposed method increases training complexity as the number of transformations increases. This maybe taxing if the transformation space is large. The transformation space used by the authors is relatively small, it would be good to get evidence if the ranking loss scales well with increase in the number of transformations. Since at runtime the allowed budget is usually in 1-5, inability to correctly rank the expected losses could lead to regression in performance in a large transformation space. Due to the nature of the proposed method, scaling to new transformations needs retraining.

Correctness: Yes.

Clarity: The paper is written clearly, however explaining the details of the ranking loss function would aid understanding. L136: It is not very clear what the authors are trying to convey.

Relation to Prior Work: Yes the authors reference recent relevant works appropriately.

Reproducibility: No

Additional Feedback:

[Author Response · NeurIPS 2020]

We would like to thank you for your thorough evaluation, helpful suggestions, and comments. We here address the key concerns and note that the paper will be updated accordingly. We conducted new experiments to reinforce the evidence for this response, as reviewers suggested. Before we begin, however, we emphasize that the paper provides the first efficient instance-aware test-time augmentation method resulting in significant gains over previous approaches.

Table 1: ImageNet(-C) result of ResNet-50 with the standard training-time augmentations.

| Test-time Augmentation | Relative Cost | Clean Test-set | Corrupted set mCE | Corrupted Test-set mCE |
|---|---|---|---|---|
| Center-Crop | 1 | 24.14 | 78.93 | 75.42 |
| Horizontal-Flip | 2 | 23.76 | 77.91 | 74.32 |
| 5-Crops | 5 | 23.91 | 77.52 | 73.87 |
| 10-Crops | 10 | 23.04 | 76.69 | 72.98 |
| Random($k=1$) | 1 | 26.89 | 82.86 | 79.81 |
| Random($k=2$) | 2 | 25.14 | 79.91 | 77.00 |
| Random($k=4$) | 4 | 24.29 | 78.24 | 75.38 |
| GPS($k=1$) | 1 | 24.86 | 82.13 | 79.43 |
| GPS($k=2$) | 2 | 23.78 | 76.45 | 73.32 |
| GPS($k=4$) | 4 | 23.44 | 77.27 | 73.87 |
| GPS†($k=1$) | 1 | 27.39 | 77.21 | 75.07 |
| GPS†($k=2$) | 2 | 27.04 | 76.48 | 74.27 |
| GPS†($k=4$) | 4 | 26.88 | 76.09 | 73.84 |
| Ours($k=1$) | 1 | 24.14 | 75.52 | 74.29 |
| Ours($k=2$) | 2 | 24.10 | 75.00 | 73.61 |
| Ours($k=2$) + Flip | 4 | 23.74 | 74.00 | 72.59 |

Figure 1: Comparison for the same 5 Crop candidates on the clean ImageNet set using ResNet-50. Top-1 accuracies by the number of ensembles. We trained our loss predictor for five crop areas. Compared to the 5-crop ensemble, choosing one transform by our method gives almost the same performance, and selecting the two transforms achieves even better performance with less computational cost.

Figure 2: Comparison for the same GPS transforms on the clean ImageNet set using ResNet-50. Top-1 accuracies by the number of ensembles. We trained our loss predictor on the searched GPS policies to choose ones specific for each test instance. Our method properly selects valid transforms from the candidates chosen greedily by GPS, and therefore further improves the performances over static ensemble from GPS.

*GPS : Greedy Policy Search on the clean dataset.
*GPS†: Greedy Policy Search on the corrupted dataset.

**Comparison study with GPS (Greedy Policy Search) [30] (R1+R2+R3):** The official code for GPS (released after our submission) is used for comparison. In Table 1, we show that the proposed method outperformed both GPS and GPS† on ImageNet-C. This means that the performances of GPS on both seen and unseen corruptions lagged behind our proposed method. In particular, the GPS policies found on the corrupted dataset produced poor results in the clean set, while our method prevented the performance degradation on the clean set. We confirmed by the GPS code that the search space of GPS includes all our augmentation policies such as "auto-contrast" and "sharpness"; our performance gains come from the proposed instance-specific transformation. A detailed comparison will be included.

**Test-time augmentation for the clean set (R2+R3):** We conducted experiments with loss predictor trained for the clean set. In Figure 1, our loss predictor picks out promising one out of five crop regions. Even if only one crop region is selected using the loss predictor, the obtained performance is comparable to the existing 5-crop ensemble. This is clear proof that our method is also effective on the clean set and separated from our search space, our loss predictor itself contributes to enhancing the classification performance. We will include the result of the clean test-set.

**Validating loss predictor (R2+R3):** Firstly, we add random baselines with $k \geq 1$. As $k$ increases, the random baselines' performance marginally increases, but our method using loss predictor instead of random selection shows significant improvements. Secondly, in Table 1, our method uses augmentation space, which is a subset of GPS's space. Nevertheless, our performance is better since we select the best one for the test instance with the loss predictor. Lastly, in Figure 1 and 2, it is also a critical rationale that performance increases consistently when the order of the policy is dynamically determined with the loss predictor. We will elaborate more on this in the revised paper.

**Details of the loss function (R1+R4):** We used the surrogated ranking loss proposed in [8], as described in L171. Specifically, to optimize the non-differentiable Spearman correlation between relative losses and predictions, we trained a recurrent neural network that approximates the correlation using the official implementation. This surrogate loss function has been chosen after an extensive comparison with others. We will revise the paper with the details.

**R1:** We will add the missing related works and add calibrated log-likelihood to our revised paper. **R2:** As the reviewer pointed out, our test-time transformations consist of basic operations that may restore a corrupt image close to normal. However, transformations for a given test image is selected by the loss predictor. As [17] shows, manually targeted image restoration can be harmful to robustness when the corruption of each test image is unknown at test-time. In addition, as shown in Table 2 and 3 in the manuscript, taking into account our transformations at training-time of the target network leads to performance degradation on some corruptions and (most importantly) clean set. The proposed loss predictor contributes to picking the most proper one that not distorts more but may restore a corrupted image, which improves the robustness of the target network in a consistent way. **R3:** In Table 1, we compare the performance of baselines and our method on the ResNet-50 trained with the standard train-time augmentation. We will update the experimental results with various train-time augmentations and more baselines, and revise the manuscript to reflect your additional comments. **R4:** As the number of transformations increases, the cost of transforming and inferencing the input linearly increases. But this is highly parallelizable. Also, in this study, we prepared a small setting to focus on demonstrating the potentials of instance-aware test-time augmentation. Although the augmentation space is limited, the experiment results show the superiority of our methods against previous approaches. Also, applying augmentations repeatedly to expand transformation space in a combinatorial way is promising in our experiment. We expect future works to be conducted in the direction of using less cost while expressing more augmentations.

[Meta-Review · NeurIPS 2020]

The work proposes a learned loss function for test-time augmentation, with sufficient baseline improvements that all reviewers after discussion found sufficiently convincing. Adding some of the clarifications in the rebuttal (e.g., loss function), and additional experiments, as revisions to the paper would significantly improve the work.